# Improving Deep Policy Gradients with Value Function Search

**Enrico Marchesini, Christopher Amato**
Khoury College of Computer Sciences
Northeastern University
Boston, MA, USA
`{e.marchesini, c.amato}@northeastern.edu`

## Abstract

Deep Policy Gradient (PG) algorithms employ value networks to drive the learning of parameterized policies and reduce the variance of the gradient estimates. However, value function approximation gets stuck in local optima and struggles to fit the actual return, limiting the variance reduction efficacy and leading policies to sub-optimal performance. This paper focuses on improving value approximation and analyzing the effects on Deep PG primitives such as value prediction, variance reduction, and correlation of gradient estimates with the true gradient. To this end, we introduce a Value Function Search that employs a population of perturbed value networks to search for a better approximation. Our framework does not require additional environment interactions, gradient computations, or ensembles, providing a computationally inexpensive approach to enhance the supervised learning task on which value networks train. Crucially, we show that improving Deep PG primitives results in improved sample efficiency and policies with higher returns using common continuous control benchmark domains.

## 1 Introduction

Deep Policy Gradient (PG) methods achieved impressive results in numerous control tasks (Haarnoja et al., 2018). However, these methods deviate from the underlying theoretical framework to compute gradients tractably. Hence, the promising performance of Deep PG algorithms suggests a lack of rigorous analysis to motivate such results. Ilyas et al. (2020) investigated the phenomena arising in practical implementations by taking a closer look at key PG primitives (e.g., gradient estimates, value predictions). Interestingly, the learned value networks used for predictions (*critics*) poorly fit the actual return. As a result, the local optima where critics get stuck limits their efficacy in the gradient estimates, driving policies (*actors*) toward sub-optimal performance.

Despite the lack of investigations to understand Deep PG's results, several approaches have been proposed to improve these methods. Ensemble learning (Lee et al., 2021; He et al., 2022), for example, combines multiple learning actors' (or critics') predictions to address overestimation and foster diversity. These methods generate different solutions that improve exploration and stabilize the training, leading to higher returns. However, the models used at training and inference time pose significant challenges that we discuss in Section 5. Nonetheless, popular Deep Reinforcement Learning (RL) algorithms (e.g., TD3 (Fujimoto et al., 2018)) use two value networks, leveraging the benefits of ensemble approaches while limiting their complexity. To address the issues of ensembles, gradient-free methods have been recently proposed (Khadka & Tumer, 2018; Marchesini & Farinelli, 2020; Sigaud, 2022). The idea is to complement Deep PG algorithms with a search mechanism that uses a population of perturbed policies to improve exploration and to find policy parameters with higher payoffs. In contrast to ensemble methods that employ multiple actors and critics, gradient-free population searches typically focus on the actors, disregarding the value network component of Deep PG. Section 5 discusses the limitations of policy search methods in detail. However, in a PG context, critics have a pivotal role in driving the policy learning process as poor value predictions lead to sub-optimal performance and higher variance in gradient estimates (Sutton & Barto, 2018).

Such issues are further exacerbated in state-of-the-art Deep PG methods as they struggle to learn good value function estimation Ilyas et al. (2020). For this reason, we propose a novel gradient-free population-based approach for critics called Value Function Search (VFS), depicted in Figure 1. We aim to improve Deep PG algorithms by enhancing value networks to achieve (i) a better fit of the actual return, (ii) a higher correlation of the gradients estimate with the (approximate) true gradient, and (iii) reduced variance. In detail, given a Deep PG agent charac-

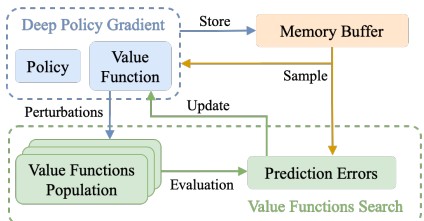

Figure 1: Overview of VFS.

terized by actor and critic networks, VFS periodically instantiates a population of perturbed critics using a two-scale perturbation noise designed to improve value predictions. Small-scale perturbations explore *local* value predictions that only slightly modify those of the original critic (similarly to gradient-based perturbations (Lehman et al., 2018; Martin H. & de Lope, 2009; Marchesini & Amato, 2022)). Big-scale perturbations search for parameters that allow escaping from local optima where value networks get stuck. In contrast to previous search methods, evaluating perturbed value networks require computing standard value error measures using samples from the agent's buffer. Hence, the Deep PG agent uses the parameters with the lowest error until the next periodical search. Crucially, VFS's critics-based design addresses the issues of prior methods as it does not require a simulator, hand-designed environment interactions, or weighted optimizations. Moreover, our population's goal is to find the weights that minimize the same objective of the Deep PG critic.

We show the effectiveness of VFS on different Deep PG baselines: (i) Proximal Policy Optimization (PPO) (Schulman et al., 2017), (ii) Deep Deterministic Policy Gradient (DDPG) (Lillicrap et al., 2016), and (iii) TD3 on a range of continuous control benchmark tasks (Brockman et al., 2016; Todorov et al., 2012). We compare over such baselines, an ensembled Deep RL method, SUNRISE (Lee et al., 2021), and the policy search Supe-RL (Marchesini et al., 2021a). In addition, we analyze key Deep PG primitives (i.e., value prediction errors, gradient estimate, and variance) to motivate the performance improvement of VFS-based algorithms. Our evaluation confirms that VFS leads to better gradient estimates with a lower variance that significantly improve sample efficiency and lead to policies with higher returns. Our analysis highlights a fundamental issue with current state-of-the-art Deep PG methods, opening the door for future research.

## 2  BACKGROUND

PG methods parameterize a policy $\pi_\theta$ with a parameters vector $\theta$ (typically the weights of a Deep Neural Network (DNN) in a Deep PG context), and $\pi_\theta(a_t|s_t)$ models the probability to take action $a_t$ in a state $s_t$ at step $t$ in the environment. These approaches aim to learn $\theta$ following the gradient of an objective $\eta_\theta$ over such parameters (Sutton & Barto, 2018). Formally, the primitive gradient estimate on which modern Deep PG algorithms build has the following form (Sutton et al., 1999):

$$\nabla\eta_\theta = \mathbb{E}_{(s_t, a_t)\in\tau\sim\pi_\theta}\left[\nabla_\theta\log\pi_\theta(a_t|s_t)Q_{\pi_\theta}(s_t, a_t)\right] \tag{1}$$

where $(s_t, a_t) \in \tau \sim \pi_\theta$ are states and actions that form the trajectories sampled from the distribution induced by $\pi_\theta$, and $Q_{\pi_\theta}(s_t, a_t)$ is the expected return after taking $a_t$ in $s_t$. However, Equation 1 suffers from high-variance expectation, and different baselines have been used to margin the issue. In particular, given a discount value $\gamma \in [0, 1)$, the state-value function $V_{\pi_\theta}(s)$ is an ideal baseline as it leaves the expectation unchanged while reducing variance (Williams, 1992):

$$V_{\pi_\theta}(s) = \mathbb{E}_{\pi_\theta}\left[G_t := \sum_t^\infty \gamma^t r_{t+1}|s_t = s\right] \tag{2}$$

where $G_t$ is the sum of future discounted rewards (*return*). Despite building on the theoretical framework of PG, Deep PG algorithms rely on several assumptions and approximations for designing feasible updates for the policy parameters. In more detail, these methods typically build on surrogate objective functions that are easier to optimize. A leading example is TRPO (Schulman et al., 2015), which ensures that a surrogate objective updates the policy locally by imposing a trust region. Formally, TRPO imposes a constraint on the Kullback–Leibler diverge (KL) on successive policies $\pi_\theta, \pi_{\theta'}$, resulting in the following optimization problem:

$$\max_\theta \mathbb{E}_{\pi_\theta}\left[\frac{\pi_\theta(a_t|s_t)}{\pi_{\theta'}(a_t|s_t)}A_{\pi_\theta}(s_t, a_t)\right] \text{ s.t. } D_{KL}(\pi_\theta(\cdot|s)||\pi_{\theta'}(\cdot|s)) \leq \delta \tag{3}$$

where $\delta$ is a hand-tuned divergence threshold, and $A(s, a) = Q(s, a) - V(s)$ is the advantage function. In practical implementations, Equation 3 is tractably optimized with additional tricks: (i) a second-order approximation of the objective, (ii) a mean KL of the current trajectory instead of the actual KL divergence. Given TRPO's computational demands, Schulman et al. (2017) further "relaxes" the optimization landscape by replacing the constraint with the following clipped objective, providing a significantly faster (i.e., less demanding) yet better-performing alternative over TRPO:[1]

$$\max_{\theta} \; \mathbb{E}_{\pi_\theta} \left[ \min \left( \frac{\pi_\theta(a_t|s_t)}{\pi_{\theta'}(a_t|s_t)} A(s_t, a_t), \text{clip}\left( \frac{\pi_\theta(a_t|s_t)}{\pi_{\theta'}(a_t|s_t)}, 1 - \epsilon, 1 + \epsilon \right) A(s_t, a_t) \right) \right] \quad (4)$$

Similarly, Lillicrap et al. (2016) uses the deterministic PG theorem (Silver et al., 2014) for learning deterministic policies $\mu_\theta(s)$. DDPG and further optimizations (e.g., TD3) are tightly related to standard Q-learning (Watkins & Dayan, 1992) and assume having a differentiable action-value function over actions. Deterministic PG algorithms learn the value parameters as in standard Double DQN. In contrast, policy parameters are learned under the $\max_\theta \; \mathbb{E}_{\pi_\theta} [Q_\phi(s, \mu_\theta(s))]$ optimization problem, where $\phi$ are the parameters of the critic. Moreover, TD3 addresses the overestimation of DDPG with additional tricks, e.g., learning two value functions, being pessimistic about the action value to use for the target computation, and delaying the policy updates.

In summary, Deep PG algorithms build their success on learning value functions for driving the learning of the policy parameters $\theta$. However, significant research efforts have been devoted to improving policies, while the attention to improving critics appears marginal. Hence, to our knowledge, VFS is the first gradient-free population-based approach that works on top of Deep PG to improve critics without facing the complexities of learning ensembles or policy searches.

## 2.1 Gradient-Informed Perturbations

Policy search approaches generate a population of perturbed versions of the agent's policy $\pi_\theta$ to explore variations with higher returns (Khadka & Tumer, 2018; Marchesini & Farinelli, 2022; Marchesini et al., 2021b; Colas et al., 2018; Sigaud, 2022). To this end, gradient-informed perturbations (or variations) have been employed for limiting detrimental effects on $\pi_\theta$, exploring small changes in the population's action sampling process (Lehman et al., 2018; Marchesini et al., 2022). In more detail, prior methods express the average divergence of a policy at state $s$ over a perturbation $\omega$ as:

$$d(\pi_\theta, \omega) = \|\pi_\theta(\cdot|s) - \pi_{\theta+\omega}(\cdot|s)\|_2 \quad (5)$$

and use the gradient of such divergence to approximate the sensitivity of $\theta$ over the policy decisions, which is used to normalize Gaussian-based perturbations. Formally, following Lehman et al. (2018):

$$\nabla_{\theta_a} d(\pi_\theta, \omega) \approx \nabla_\theta d(\pi_\theta, 0) + H_\theta(d(\pi_\theta, 0))\omega \quad (6)$$

where $H_\theta$ is the Hessian of divergence with respect to $\theta$. Although simpler first-order approximations achieve comparable results, gradient-based perturbations add significant overhead to the training process. Prior work introduced gradient-based variations because evaluating the population against detrimental changes is hard. As such, the quality of perturbed policies is assessed over many trajectories to ensure that they effectively achieve higher returns over arbitrary initializations (Marchesini et al., 2021a). In contrast, we focus on a critics-based search where high-scale perturbations may help the learning process to escape from local optima, achieving better predictions. Hence, VFS relies on a two-scale perturbation approach to maintain similar value predictions to the original value function while exploring diverse parameters to escape from local optima.

## 3 Value Function Search

We introduce a gradient-free population-based search with two-scale perturbations to enhance critics employed in Deep PG. Ultimately, we aim to show that value functions with better predictions improve Deep PG primitives, leading to better sample efficiency and policies with higher returns.

Algorithm 1 shows the general flow of the proposed Value Function Search framework. During a standard training procedure of a Deep PG agent, VFS periodically generates a population $\mathcal{P}$ of perturbed value networks, starting from the agent's current critic parametrized by $\phi$. We first instantiate

---

[1]We do not consider the unconstrained penalty-based PPO due to the better performance of the clipped one.

$n$ copies of $\phi$ and two Gaussian noise distributions $\mathcal{G}_{\min,\max}$ with $\sigma_{\min,\max}$ standard deviation for the two-scale variations (lines 3-4). Hence, we sample from the two distributions to apply each perturbation to half of the population (lines 5-6). In contrast to single-scale or gradient-based perturbations of prior work (Khadka & Tumer, 2018; Marchesini et al., 2021a; Sigaud, 2022), the two scales approach for value networks has the following advantages:[2] (i) $\mathcal{G}_{\min}$ maintains similar predictions to those of the unperturbed critic, similarly to gradient-based variations. Sampling from $\mathcal{N}(0, \sigma_{\min})$ translates into small changes in the value prediction process to find a better set of parameters within a local search fashion. (ii) $\mathcal{G}_{\max}$ is a Gaussian with a greater standard deviation $\mathcal{N}(0, \sigma_{\max})$ used to explore diverse value predictions to escape from local optima. (iii) Gaussian-based perturbations do not introduce significant overhead as gradient-based variations. After generating the population, VFS samples a batch $b$ of trajectories from the agent's buffer $B$. There are different strategies for sampling the required experiences from the agent's memory based on the nature of the Deep PG algorithm. When using an on-policy agent (e.g., TRPO, PPO), we apply VFS after updating the policy and value networks. In this way, we reuse the same on-policy data to evaluate whether a perturbed set of weights has better value prediction (i.e., lower error). Off-policy agents (e.g., DDPG, TD3) could follow a similar strategy. Still, the reduced size of the mini-batches typically employed by off-policy updates would result in noisy evaluations. For this reason, we sample a larger batch of experiences to provide a more robust evaluation. In practice, to compute the prediction errors for each value network in the population (line 7), we use the typical objective used to train PG critics, the Mean Squared Error (MSE) (Sutton & Barto, 2018), whose general formulation is:

$$mse_{\phi_i}(b) = \frac{1}{|b|} \sum_{s \in b} \left[ V_\pi(s) - V_{\pi_{\phi_i}}(s) \right]^2 \tag{7}$$

where $V_\pi(s)$ is the true value function. However, it is typically unfeasible to access the true $V_\pi(s)$ and, as in standard training procedures of Deep PG algorithms, we rely on noisy samples of $V_\pi(s)$ or bootstrapped approximations. In more detail, on-policy updates are particularly interesting, as Deep PG algorithms typically learn critics with Monte-Carlo methods without bootstrapping the target value. Hence, performing a supervised learning procedure on $(s_t, G_t)$ pairs, where the return $G_t$ is an unbiased, noisy sample of $V_\pi(s)$ that replaces the expectation of Equation 2 with an empirical average. We then refer to a perturbed critic as a *local* improvement of the original unperturbed one when it achieves a lower error on $b$. In our context, given a value network parametrized by $\phi$, a batch of visited trajectories $b$, and a perturbation $\omega$, we define the value network parametrized by $\phi + \omega$ to be a local improvement of $\phi$ when $mse_{\phi+\omega}(b) \leq mse_\phi(b)$. Moreover, in the limit where $b$ samples all the trajectories, $G_t \equiv V_\pi(s_t)$, and $mse_{\phi+\omega}(b) \leq mse_\phi(b)$ means that $\phi + \omega$ is a better or equal fit of the actual value function with respect to the original $\phi$. Crucially, Monte-Carlo value approximation converges to a local optimum even when using non-linear approximation (Sutton & Barto, 2018). Hence, on-policy VFS potentially maintains the same convergence guarantees of the baseline.[3] To this end, it would be possible to anneal the perturbation scales to zero out the impact of VFS over the training phase. Hence, providing the diversity benefits of VFS early in the training while maintaining the convergence guarantee of the baseline algorithm within the limit.

---

**Algorithm 1** Value Function Search

---

1: **Given:** (i) a Deep PG agent with a value network parametrized by $\phi$ at training epoch $e$; (ii) a buffer $B$ of visited trajectories; (iii) periodicity $e_s$ for VFS and population size $n$; (iv) scale $\sigma_{\min,\max}$ for the two-scale Gaussian noise $\mathcal{G}$.
2: **if** $e \% e_s = 0$ **then**
3:     $\mathcal{P} \leftarrow \{\phi_0, \ldots, \phi_n\}$ copies of $\phi$
4:     $\mathcal{G}_{\min,\max} \leftarrow \mathcal{N}(0, \sigma_{\min,\max})$
5:     $\phi_{1:\frac{|P|}{2}} \leftarrow \phi_i + \mathcal{G}_{\min} \quad \phi_{\frac{|P|}{2}:n} \leftarrow \phi_i + \mathcal{G}_{\max}$
6:     $b \leftarrow$ Sample a batch of trajectories from $B$
7:     $mse_\mathcal{P} \leftarrow$ Evaluate $\mathcal{P}$ over $b$ as Equation 7
8:     $\phi \leftarrow \min\limits_{\phi_i \in \mathcal{P}}(mse_\mathcal{P})$
9: **end if**

---

[2]We support claims on our perturbation approach with additional experiments in Section 4.

[3]Due to bootstrapping, the same result generally does not apply to temporal difference targets.

Nonetheless, as discussed in Ilyas et al. (2020), Deep PG algorithms deviate from the underlying theoretical framework. We performed additional experiments in Section 4 to show that maintaining VFS through the entire learning process results in better performance over zeroing out the perturbations. Moreover, in a worst-case scenario where the population search never improves the value prediction locally, VFS matches the same local optima of the original approach. However, following the insights of Fujimoto et al. (2018), practical Deep PG algorithms can not theoretically guarantee to reduce estimation errors or improve predictions outside the batch used for computing the update.

VFS addresses most limitations of previous policy search approaches and minimizes the same objective of the Deep PG critics, leading to the following crucial advantages:

- VFS does not require additional environment interactions or hand-designed evaluations for selecting the best set of parameters in the population.
- MSE is a widely adopted objective for learning value networks, providing an established evaluation strategy. Moreover, batch computations and parallelization make this process particularly efficient. Nonetheless, VFS is not explicitly designed for MSE, and extending it to different error measures is straightforward.
- At the cost of adding overhead, the population evaluation may scale to an arbitrarily large number of samples to improve robustness.

Finally, the parameters with the lowest error replace the current agent's critic until the next population search (line 8). Many Deep RL algorithms employ target networks parametrized by $\phi_{tg}$ to reduce overestimation. In this scenario, we also update the target weights toward the new ones by using standard Polyak averaging with an $\alpha \in (0, 1)$ weight transfer value: $\phi_{tg} = \alpha\phi + (1 - \alpha)\phi_{tg}$.

**Limitations.** Here we highlight the limitations of VFS: (i) batched computations and parallelization do not avoid a marginal overhead induced by the population search. (ii) In an off-policy setting, VFS requires tuning the batch size for the error computation to balance the trade-off between computational demands and the evaluation quality. (iii) There are no guarantees that small-scale perturbations produce a local variation to the original predictions. However, they have been previously shown effective in doing so (Martin H. & de Lope, 2009), and we confirm this result in Section 4.

## 4 EXPERIMENTS

The proposed experiments aim at answering the following questions: (i) *How do VFS components affect the training phase*. In particular, we investigate when the perturbations are most effective during the training. (ii) *How these components impact the performance and sample efficiency* of VFS-based algorithms. (iii) *How VFS influences Deep PG primitives, such as gradient estimates, their variance, and value predictions*, to motivate performance improvement.

We investigate the performance of VFS on the on-policy PPO, the off-policy DDPG, and TD3 algorithms. We compare over the baseline DDPG, PPO, TD3, SUNRISE (Lee et al., 2021) (a recent ensemble approach), and Supe-RL (Marchesini et al., 2021a) (a recent policy search approach). We conduct our experiments on five continuous control tasks based on MuJoCo (Brockman et al., 2016) (in their v4 version), which are widely used for comparing Deep PG, ensembles, and policy search approach (Schulman et al., 2017; Lillicrap et al., 2016; Lee et al., 2021; Fujimoto et al., 2018).

Data are collected on nodes equipped with Xeon E5-2650 CPUs and 64GB of RAM, using the hyperparameters of Appendix E. Considering the importance of having statistically significant results (Colas et al., 2019), we report the average performance of 25 runs per method, with shaded regions representing the standard error. Such a high number of experiments motivates slightly different results over the original implementations of DDPG, TD3, and PPO.

### 4.1 ANALYZING VFS COMPONENTS

This analysis considers data collected in the Hopper task with VFS-PPO, as we achieved comparable results in the other setups. Given our intuitions, the two-scale perturbations have different effects on the critic. We expect $\sigma_{\min}$ perturbation to result in networks with similar predictions. In contrast, $\sigma_{\max}$ should generate networks with more diversified values to escape local optima. Appendix A shows the average difference between values predicted by the original critic of the baselines and their variations, confirming the role of the two perturbations.

Table 1: Performance at 1M steps for Hopper, Walker, Swimmer, and 2M steps for HalfCheetah, and Ant in PPO, DDPG, TD3 experiments. We show the average return and standard deviation over the 25 runs and the percentage of steps required by VFS methods to reach the peak performance of the corresponding baseline. We report the best number for the ensemble SUNRISE at $2 \times 10^5$ steps (Lee et al., 2021), comparing over a SAC implementation of VFS and the policy search Supe-RL.

| | PPO | VFS-PPO | | | DDPG | VFS-DDPG | |
|---|---|---|---|---|---|---|---|
| | *Avg. Return* | | *Steps %* | | *Avg. Return* | | *Steps %* |
| **Hopper** | $2287 \pm 805$ | $\mathbf{2878 \pm 695}$ | -48.5 | **Hopper** | $1734 \pm 810$ | $\mathbf{2397 \pm 637}$ | -34.9 |
| **Walker** | $2384 \pm 1074$ | $\mathbf{2905 \pm 932}$ | -12.4 | **Walker** | $1753 \pm 858$ | $\mathbf{2492 \pm 755}$ | -38.2 |
| **Swimmer** | $80 \pm 26$ | $\mathbf{86 \pm 22}$ | -38.6 | **Swimmer** | $46 \pm 2$ | $\mathbf{49 \pm 1}$ | -36.1 |
| **HalfCheetah** | $2560 \pm 767$ | $\mathbf{3471 \pm 252}$ | -38.8 | **HalfCheetah** | $7948 \pm 2702$ | $\mathbf{9208 \pm 711}$ | -45.4 |
| **Ant** | $4333 \pm 677$ | $\mathbf{5403 \pm 120}$ | -27.1 | **Ant** | $1060 \pm 938$ | $\mathbf{2060 \pm 446}$ | -24.4 |

| | TD3 | VFS-TD3 | | | VFS-SAC | Supe-RL (SAC) | SUNRISE |
|---|---|---|---|---|---|---|---|
| | *Avg. Return* | | *Steps %* | | *Avg. Return* | | |
| **Hopper** | $2558 \pm 1029$ | $\mathbf{3185 \pm 398}$ | -44.8 | **Hopper** | $\mathbf{2573 \pm 420}$ | $992 \pm 360$ | $2643 \pm 472$ |
| **Walker** | $3773 \pm 1052$ | $\mathbf{4185 \pm 860}$ | -58.36 | **Walker** | $\mathbf{2101 \pm 798}$ | $1932 \pm 842$ | $1236 \pm 1123$ |
| **Swimmer** | $123 \pm 27$ | $\mathbf{134 \pm 11}$ | -34.0 - | **HalfCheetah** | $7145 \pm 822$ | $\mathbf{7177 \pm 1226}$ | $4501 \pm 443$ |
| **HalfCheetah** | $10232 \pm 1201$ | $\mathbf{12430 \pm 551}$ | -58.3 | **Ant** | $\mathbf{1667 \pm 481}$ | $1398 \pm 676$ | $1502 \pm 483$ |
| **Ant** | $4003 \pm 727$ | $\mathbf{4669 \pm 382}$ | -20.4 | | | | |

Moreover, we expect $\sigma_{\max}$ variations to be rarely better than $\sigma_{\min}$ variations and the original critic, as drastic changes in the value predictions serve to escape from local optima. To confirm this, Figure 2 shows the average cumulative number of times on which $\sigma_{\min}$ (blue) or $\sigma_{\max}$ (orange) critics' error is the lowest in the population compared to the original critic (y-axis), over the periodical searches (x-axis). The linear growth of the $\sigma_{\min}$ bars indicates that such perturbations produce better variations over $\sigma_{\max}$ and the original critic through the training. In contrast, $\sigma_{\max}$ perturbed networks have more impact in the early stages of the training (where the critics' predictions are seldom accurate) and occasionally achieve lower error in later stages of the training.

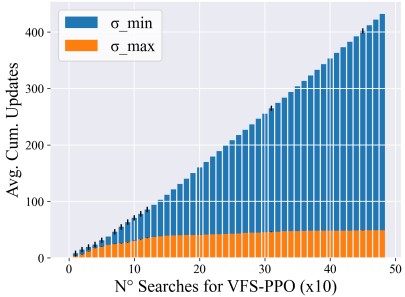

Figure 2: Cumulative number of times when a $\sigma_{\min,\max}$ variation improve the critic and the other perturbation.

## 4.2 EMPIRICAL EVALUATION

Here we discuss the performance and sample efficiency improvement of VFS-based algorithms. In particular, we show: (i) the average return at convergence and (ii) the percentage of samples required by VFS implementations to achieve the peak performance of the Deep PG baselines. Table 1 shows the results for PPO, VFS-PPO, DDPG, VFS-DDPG, TD3, and VFS-TD3 (the VFS-TD3 algorithm applies the search to both critics). In each table, we highlight the highest average return in bold and refer to Appendix B for an exhaustive overview of the training curves. Generally, VFS-based algorithms significantly improve average return and sample efficiency, confirming the benefits of enhancing critics during Deep PG training procedures. On average, VFS-PPO achieves a 26% return improvement and uses 33% fewer samples to reach the peak performance of PPO. VFS-DDPG has a 29% return improvement and 35% improvement in sample efficiency. Moreover, VFS-TD3 achieves an average 19% return improvement and 43% fewer samples to reach the peak performance of TD3. The latter results are significant as TD3 employs several tricks to address the poor performance of DDPG, and it is known to be a robust Deep PG baseline (Fujimoto et al., 2018).

We also compare VFS over Supe-RL and the ensemble method SUNRISE by considering its best number reported at $2 \times 10^5$ steps. We use the same population parameters as in our VFS-TD3 implementation and Supe-RL, while the SAC parameters are the same as the original work (Haarnoja et al., 2018).[4] Crucially, the VFS implementation shows comparable or superior average performance over such additional baselines ($\approx 13\%$ and $\approx 24\%$ on average, respectively), confirming the merits of our framework. Finally, Appendix C contains: (i) a comparison of the computational

---

[4]We do not consider the Swimmer environment as prior work has not employed it.

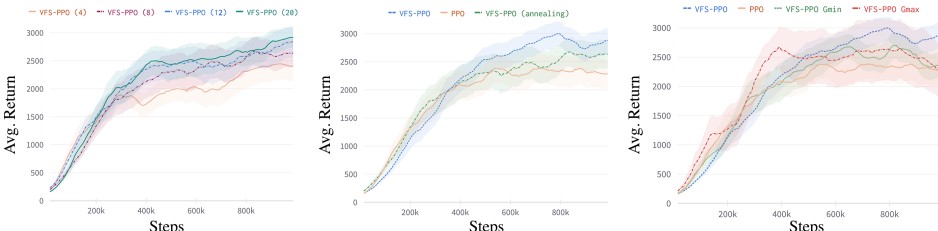

Figure 3: Left: influence of different population sizes in VFS-PPO. Increasing the size of the population leads to higher performance. Middle: comparison between PPO, VFS-PPO, and VFS-PPO with annealing perturbations. Linearly scaling our perturbations to zero leads to lower performance with respect to the proposed VFS-PPO with fixed permutations. Right: comparison of two VFS-PPO implementations that consider only small or big scale variance for the permutations.

overhead for VFS, Supe-RL, and SUNRISE, discussing the negligible overhead of VFS over such baselines. (ii) A comparison with a baseline PPO that considers a higher number of gradient steps to match the computational demands of VFS.

### 4.2.1 ADDITIONAL EXPERIMENTS

This section shows: (i) the effects of different population sizes; (ii) additional experiments to show that annealing variations have lower performance than the proposed VFS; (iii) how VFS behaves with considering only $\sigma_{\min}$ or $\sigma_{\max}$ perturbations. Similarly to Section 4.1, we show the results of VFS-PPO in the Hopper domain as they are also representative of the other setups.

Figure 3 on the left shows the performance with increasing sizes for the population (i.e., 4, 8, 12, 20). Intuitively, increasing the size of the population leads to higher performance. However, our experiments show no significant advantage in considering a population of size greater than 10.

In the center, Figure 3 compares PPO, VFS-PPO, and VFS-PPO with annealing on the perturbations. In more detail, we linearly scale the effects of the perturbations to zero towards step $9 \times 10^5$. As discussed in Section 3, after zeroing out VFS's perturbations, our approach is equivalent to the considered Deep PG baseline and, as such, has the same convergence guarantee in the limit. Nonetheless, in practice, VFS-PPO without permutation annealing achieves superior performance as the critic's population search keeps improving the Deep PG primitives.

Figure 3 on the right show the performance of VFS with only small or big-scale perturbations. The latter permutations have more impact in the early stages of the training and allow for higher returns over both PPO and VFS-PPO. However, such benefits rapidly fade as the training progresses, resulting in performance similar to the PPO baseline. Conversely, small-scale variations achieve higher returns in the middle stages of the training. However, the lack of big variations to escape from local optima only leads to marginally superior performance over PPO. In contrast, the proposed two-scale VFS-PPO combines the benefits of both permutations. Such results are further motivated by an additional analysis of the gradient estimate in Appendix D.

### 4.3 DEEP PG PRIMITIVES ANALYSIS

We analyze how VFS influences Deep PG primitives to motivate performance improvement. To this end, Figure 4 left and middle show the quality of the gradient estimates and their variance at the initial and middle stages of the training (respectively), and the mean square error of PPO's critic and VFS-PPO one. Regarding gradient estimates, PPO's common implementations estimate the gradient using an empirical mean of $\approx 20^3$ samples (marked in the plots with a vertical line). Since the Deep PG agent successfully solves the task attaining a high reward, these sample-based estimates are sufficient to drive successful learning of the policy parameters.

We measure the average pairwise cosine similarity and standard deviation between 30 gradient measurements taken from the PPO and VFS-PPO models after $10^5$ and $5 \times 10^5$ steps in the Hopper task with an increasing number of state-action pairs. A cosine similarity $\in (0, 1]$ represents a positive similarity (where 0 means that the vectors are orthogonal and 1 means that the vectors are paral-

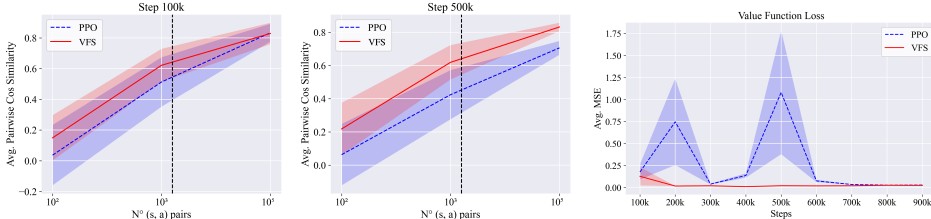

Figure 4: Left, Middle: analysis of gradient estimate and variance. We measure the average pairwise cosine similarity (y-axis) between 30 gradient estimates with a growing number of state-action pairs (x-axis). Each algorithm averages over 25 models collected during the training at $10^5$ and $5 \times 10^5$ steps. A higher correlation with a lower variance of VFS-PPO translates into better performance of the baseline (Figure 1). Right: average MSE of such models over the training phase.

lel). Hence, higher correlations indicate the gradient estimate is more similar to the actual gradient.[5] Following the PG's underlying theoretical framework, we also know that a lower variance in the gradient estimate typically translates into better performance, which is why value networks are employed in Deep PG. Crucially, VFS-based approaches show, on average, a $\approx 0.1$ higher correlation and lower variance in the sampling regime used during the training. Following the intuitions of the cosine similarity measure, such an improvement is significant and motivates the performance benefits of VFS-based approaches. We also performed a two-sample t-test to test whether the mean gradient estimates of each baseline and its VFS implementation are statistically different. In more detail, we considered the mean and standard deviation of the estimates at the sample regimes used for the algorithms (i.e., 2048 for PPO) and our sample size of 30 trials. To summarize, we can reject the equality of the average gradient estimates between PPO and VFS-PPO in the $10^5$ and $5 \times 10^5$ experiments with 95% and 99.9% confidence levels, respectively (t-stat 2.22 and 5.05).

Moreover, Figure 4 on the right shows the average MSE of the 25 trained models for PPO and VFS-PPO computed every $10^5$ step. To assess the quality of the critics' predictions, we compute the average MSE between PPO and VFS-PPO critics and the actual return of $10^3$ complete trajectories in the environment (i.e., we compute the average MSE over $10^3$ evaluation episodes every $10^5$ step). To reduce the scale of the plot and obtain a better visualization, we normalize the resulting MSE by the number of considered steps. Our results show that critics enhanced with VFS consistently result in lower MSE, which leads to the previously discussed improvements in the gradient estimates.[6] Hence, our analysis of key Deep PG primitives motivates the performance improvement of VFS-based algorithms. Appendix D shows the same analysis for VFS-DDPG and VFS-TD3, confirming our results.

## 5   RELATED WORK

Ensemble methods have been used to improve policy performance in Deep RL (He et al., 2022; Lee et al., 2021; Yang et al., 2022; Wu & Li, 2020; Ren et al., 2021) combining the predictions of multiple learning models. However, the benefits of these approaches over population-based ones are unclear, as the numerous training actors (and/or critics) introduce a significant memory footprint and non-negligible overhead at training and inference time. In addition, weighting the importance of different predictions is not straightforward. Hence, ensemble algorithms in Deep RL are unsuitable for: (i) deployment on-device where memory is limited, (ii) real-time systems where inference time is crucial, or (iii) training high-dimensional DNNs where parallel computation may not be possible. In contrast, population-based approaches are straightforward to implement and have been used to combine the exploration benefits of gradient-free methods and gradient-based algorithms (Khadka & Tumer, 2018; Marchesini et al., 2021a; Sigaud, 2022) for policy parameters search. In more detail, ERL (Khadka & Tumer, 2018) employs a Deep PG agent and a concurrent population that includes the gradient-based agent to explore different policy permutations. The agent trains following its

---

[5]Although obtaining an empirical estimate of the true gradient can improve Deep PG performance further, this would require orders of magnitude more samples than the ones used in practical applications. Using such an amount of samples is unfeasible due to the computational demands (Ilyas et al., 2020).

[6]Using more trajectories for approximating the true value function would lead to a better approximation and higher MSE for both approaches in the early stages of the training where predictions are less accurate.

gradient-based algorithm and shares a memory buffer with the population to learn from more diversified samples. However, ERL's dropout-like permutations bias the population's performance in the long run. Inspired by ERL, many policy search approaches emerged in the literature (Colas et al., 2018; Bodnar, 2020; Aloïs Pourchot, 2019; Marchesini et al., 2021a). In more detail, GEP-PG (Colas et al., 2018) uses a curiosity-driven approach to enhance the experiences in the agent's memory buffer. Moreover, Proximal Distilled ERL (PDERL) (Bodnar, 2020) introduces novel gradient-based permutations to address destructive behaviors of biologically-inspired perturbations that could cause catastrophic forgetting (Lehman et al., 2018). Finally, Super-RL (Marchesini et al., 2021a) combines the intuitions of the previous population-based approaches, proposing a framework applicable to (possibly) any DRL algorithms that do not exhibit detrimental policy behaviors. However, due to their policy-oriented nature, these approaches assume a simulated environment to evaluate the perturbed policies, which increases the sample inefficiency of Deep RL algorithms. Such additional interactions are also task-specific as they must cover a broad spectrum of behaviors to ensure a robust evaluation. As discussed in Marchesini et al. (2021a), such behavior diversity is crucial to ensure that a perturbed policy improves the original actor, not only in particular environment initializations. In addition, current policy search methods introduce gradient-based perturbation (that has a non-negligible overhead, as further discussed in Section 2) to ensure that the population's predictions do not drastically diverge from the original policy. To summarize, population-based searches address the issues of ensemble methods by exploring different behaviors for the policies but disregarding value networks. In addition, recent work (Lyle et al., 2022) discussed the capacity loss of agents that have to adapt to new prediction problems when discovering new sources of payoffs. As such, Deep PG's critics have to adjust their predictions rapidly through the training. These issues further motivate the importance of applying VFS in the training routines.

## 6 DISCUSSION

Ilyas et al. (2020) recently showed that critics employed in Deep PG algorithms for value predictions poorly fit the actual return, limiting their efficacy in reducing gradient estimate variance and driving policies toward sub-optimal performance (Fujimoto et al., 2018). Despite improving Deep PG performance, population-based search methods have recently superseded ensemble learning due to their computational benefits. However, these approaches typically need more practical investigations to understand their improvement. Moreover, previous population search frameworks have been devoted to exploring policy perturbations, disregarding critics and their crucial role in Deep PG.

To our knowledge, VFS is the first gradient-free population search approach to enhance critics and improve performance and sample efficiency. Crucially, VFS can be applied to (possibly) any Deep PG algorithm and aims at finding a set of weights that minimize the same critic's objective. Moreover, our detailed analysis of PG primitives in Section 4.3 motivates the performance improvement of the gradient-based agent due to VFS. In addition, our analysis suggests that the gap between the underlying theoretical framework of Deep PG and the actual tricks driving their performance may be less significant than previously discussed Ilyas et al. (2020). Hence, VFS opens up several exciting research directions, such as investigating the theoretical relationships between RL and practical Deep RL implementation. Empirically, VFS also paves the way for novel combinations with policy search methods to provide a unified framework of gradient-free population-based approaches. Finally, given the wide application of value networks in Deep Multi-Agent RL Rashid et al. (2018); Marchesini & Farinelli (2021); Wang et al. (2021), it would be interesting to analyze the performance improvement of VFS in such partially-observable contexts.

### 6.1 CONCLUSION

VFS introduces a two-scale perturbation operator voted to diversify a population of value networks to (i) explore local variations of current critics' predictions and (ii) allow to explore diversified value functions to escape from local optima. The practical results of such components have been investigated with additional experiments that also motivate the improvement in sample efficiency and performance of VFS-based algorithms in a range of standard continuous control benchmarks. Our findings suggest that improving fundamental Deep PG primitives translates into higher-performing policies and better sample efficiency.

## 7 ACKNOWLEDGEMENTS

This work was partially funded by the Army Research Office under award number W911NF20-1-0265.

## 8 ETHICS STATEMENT

The proposed Value Function Search does not focus on experiments involving human subjects and does not require sensitive data that involve sex, gender, and diversity analysis. Hence, we do not consider the gender dimension, privacy, data governance issues, or any primary ethic concern highlighted in the *ICLR Code of Ethics*[7] to be relevant.

However, according to the *"Ethics Guidelines for Trustworthy Artificial Intelligence"* report published by the High-Level Expert Group on AI of the European Commission,[8] ethical consideration has to take into account the environment and the sustainability of AI systems. Regarding the environmental impact, it is crucial to foster sample efficiency (i.e., reducing the training time for the agents, hence the computational resources used to train them) of Deep RL to lower energy consumption and related emissions. In this direction, VFS leads to significant benefits in sample efficiency, which reduces the environmental footprint of VFS-based approaches by a considerable margin.

In more detail, we employed the Machine Learning CO2 impact calculator (Lacoste et al., 2019) to estimate the CO2 emissions of our experiments.[9] According to our hardware, the carbon efficiency of the provider, and the length of our experiments, we computed a total emission of $\approx 52.3$ kg CO2. We plan to cope with such emissions using Treedom[10], an online service that allows us to plant trees to cope with our carbon footprint.

## 9 REPRODUCIBILITY STATEMENT

Significant efforts have been made to ensure the reproducibility of VFS. Section 3 detail the flow of VFS-based algorithms with pseudocode and detailed explanations. We also highlight possible limitations of our approach in Section 3.1. Moreover, Section 4 states our experimental setup, while supplemental material contains a detailed overview of the considered hyperparameters.

---

[7]https://iclr.cc/public/CodeOfEthics

[8]https://digital-strategy.ec.europa.eu/en/library/ethics-guidelines-trustworthy-ai

[9]We note that this is not a precise measurement as it is not trivial to map energy consumption to CO2 emissions (i.e., the carbon footprint). To this end, we employed open-source electricity maps (e.g., https://www.electricitymaps.com) to estimate better how carbon-intensive our experiment's electricity is.

[10]https://www.treedom.net

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

## A    EFFECTS OF PERTURBATIONS ON VFS-PPO, VFS-DDPG, VFS-TD3

This section shows the efficacy of using $\sigma_{\min,\max}$ to maintain similar predictions and explore diversified ones, respectively. In this direction, Figure 5 shows the average difference between values predicted by the original critic of the PPO algorithm and its two perturbations ($\sigma_{\min} = 0.000005$, $\sigma_{\max} = 0.0005$) over 100 episodes (i.e., each box is an episode). In particular, we report the data collected in the initial stage of the training (i.e., at step $10^5$), where green boxes indicate slight variations while red ones represent the highest difference in our evaluation. In more detail, Figure 5 shows the effects of our perturbations at different stages of the training (i.e., at 100k, 500k, 1M steps) for VFS-PPO, VFS-DDPG, VFS-TD3.

The values of the $\sigma_{\min}$ perturbed network differ from the original critic by a small margin. In contrast, the critic perturbed with $\sigma_{\max}$ achieves significantly different values. Hence, the proposed two-scale perturbation approach leads to the desired outcome, and $\sigma_{\min}$-based variations confirm similar results over computationally demanding gradient-based perturbations. Interestingly, our experiments with different algorithms consider the same set of hyperparameters for the mutation operators (see Appendix E), suggesting that VFS does not require extensive tuning to enhance existing Deep PG algorithms.

## B    TRAINING CURVES

Following the results in Section 4.2, Figure 6 shows the average return during the training without the five best and worst performing model.[11] We report the results for each environment in the different rows, while each column contains a different set of algorithms (i.e., VFS-PPO and PPO in the first column, VFS-DDPG and DDPG in the second column, VFS-TD3 and TD3 in the last one). As discussed in the main paper, VFS-based algorithms achieve, on average, higher returns. Interestingly, the worst-trained model for VFS-based approaches typically achieves higher returns than the best model of the baselines. In addition, VFS-based methods show a lower standard deviation, as reported in Table 1 of the main paper. Moreover, they significantly improve sample efficiency, reaching the peak performance of the baseline algorithms (i.e., PPO, DDPG, TD3) in a fraction of the steps.

We performed additional experiments in the Humanoid task for 2 million steps. We aim to show that population-based approaches can scale to more challenging tasks maintaining the same population parameters. Figure 7 shows the performance of PPO and VFS-PPO. Interestingly, we note that VFS leads to significantly higher results in this complex domain, suggesting that the limitations of critics in Deep PG algorithms profoundly impact the overall policy return.

## C    COMPUTATION OVERHEAD

In this section, we discuss the computation overhead of ensemble methods and population-based ones. In particular, we compare the overhead of training and inference time introduced by Supe-RL, SUNRISE, and VFS, considering $M = 10$ models. The training overhead is comparable for all the approaches as standard Deep RL benchmarks do not require high-dimensional DNNs or computational demanding architectures that hinder parallelization. Nonetheless, in more complex tasks requiring high-dimensional DNNs, it could be computationally unfeasible to parallelize the training models of ensemble methods, especially during backpropagation. As reported by the Supe-RL's authors (Marchesini et al., 2021a), the additional environments interactions of policy search methods lead to a maximum 6% overhead even considering parallelization with a population of size $M$. In contrast, VFS does not require multiple training models or additional environment interactions, leading to a negligible overhead for the training procedure (i.e., it simply requires additional forward DNNs propagations).

Regarding inference time, policy and value network searches do not involve multiple predictions (except in the periodical search, which has negligible overhead due to batched computations), maintaining the same inference time of the original Deep PG baselines. In contrast, as detailed by SUN-

---

[11]We removed the best and worst seeds to avoid considering the ones that lead to particularly high or low performance. The same consideration holds for Figure 3, 8.

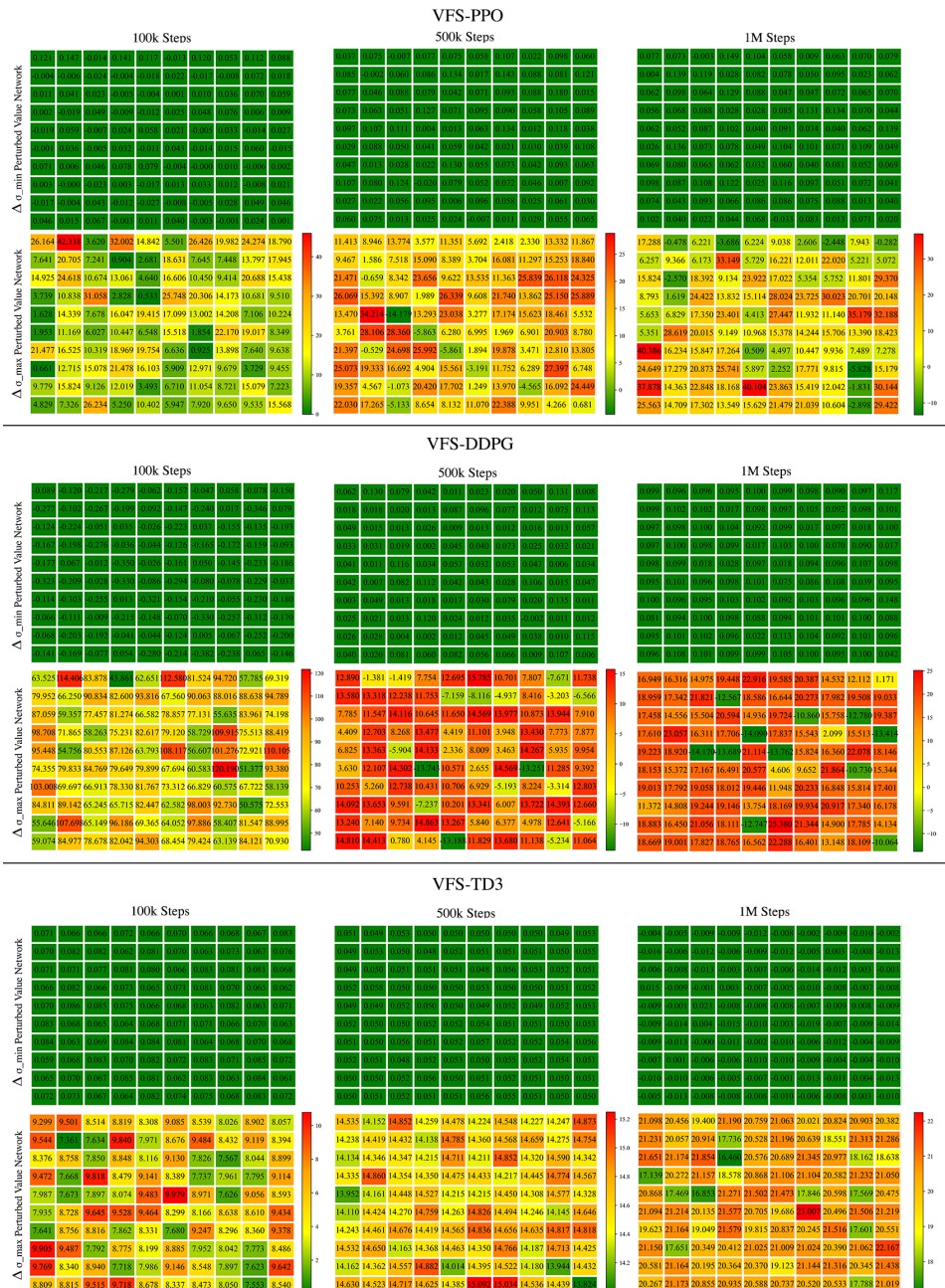

Figure 5: Efficacy of the $\sigma_{\min,\max}$ scales in modifying PPO (on the first row), DDPG (on the second row), and TD3 (on the third row) critics to achieve similar predictions or more diversified ones to escape local optima, respectively. We observe the average difference ($\Delta$) between values predicted by the critics and the $\sigma_{\min,\max}$ variations over 100 episodes in the Hopper environment shown as different boxes.

RISE's authors (Lee et al., 2021), their ensemble agents require up to $2M\times$ additional inference time.

To summarize, gradient-free search methods are generally less computationally demanding than ensemble ones. Hence, the superior return of VFS, sample efficiency, and computational benefits further confirm the merits of introducing a critics' search-based approach.

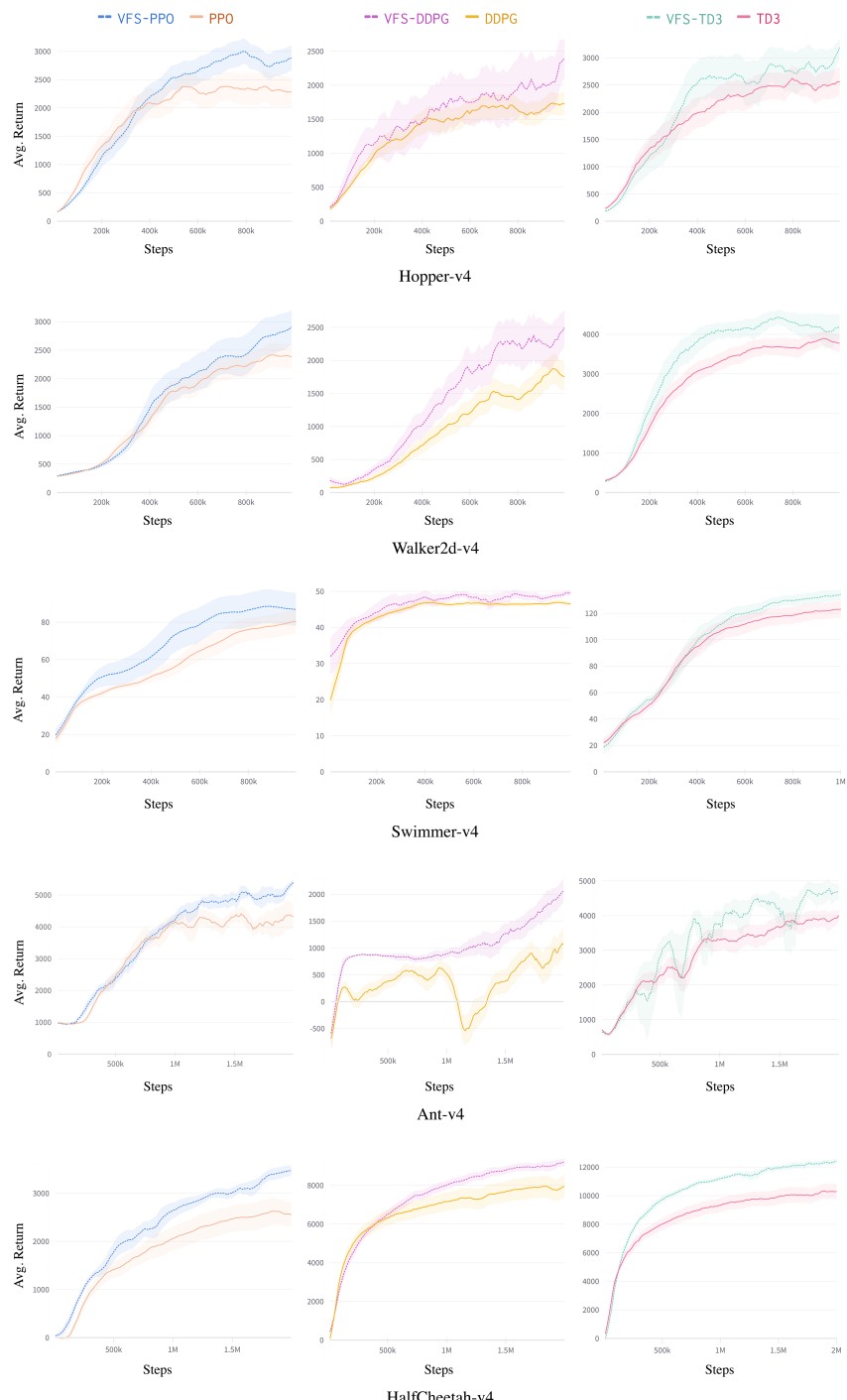

Figure 6: Comparison of PPO, DDPG, TD3, and their VFS implementation in common continuous control benchmarks on each row. Each column (i.e., each couple of algorithms) shows the average return during the training over 25 runs. VFS-based implementations improve average return and sample efficiency of the baseline Deep PG algorithms.

In addition, Figure 8 shows the performance of a PPO implementation that uses the size of the VFS population as additional gradient steps for its critic. Performing more gradient steps on the critic does not lead to significant advantages over the baseline PPO as the critic learns to fit better

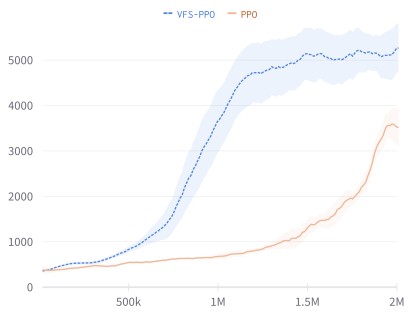

Figure 7: Comparison of PPO its VFS implementation in the Humanoid task.

the sampled batch. In contrast, VFS is designed to explore local and global variations of the critic to avoid local optima where gradient-based learning typically gets stuck. For these reasons, this additional evaluation further confirms the merits of VFS.

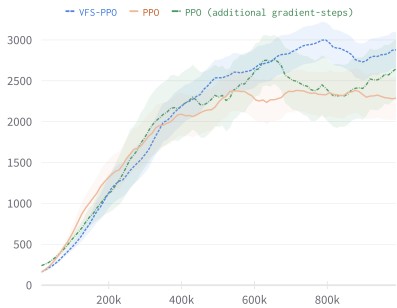

Figure 8: Comparison between VFS-PPO, PPO, and a PPO implementation that uses the size of the population as additional gradient steps for the critic.

## D    ADDITIONAL DEEP PG PRIMITIVES ANALYSIS

In Section 4.3, we analyzed the improvement of VFS-PPO in terms of gradient estimate, their variance, and the mean squared error of the critic. Such an analysis serves to motivate the performance improvement of VFS-based algorithms. In this section, we report the same comparison for VFS-DDPG and VFS-TD3 to confirm the proposed framework's benefits further.

Figure 9 reports the same results for our Deep PG primitives analysis for DDPG and TD3. We measure the average pairwise cosine similarity and its standard deviation between 30 gradient measurements taken from the algorithms at $10^5$ and $5 \times 10^5$ steps in the Hopper task with an increasing number of state-action pairs. Crucially, VFS-DDPG and VFS-TD3 show a higher correlation and lower variance in the typical sampling regime used during the training (i.e., mini-batches of size 64 or 128), motivating their performance improvement over the baselines. Similarly to the PPO case, we performed a two-sample t-test to test whether the mean gradient estimates of each baseline and its VFS implementation are statistically different. To summarize, we can reject the equality of the average gradient estimates between DDPG and VFS-DDPG with a 95% confidence level in the $10^5$ and $5 \times 10^5$ experiments (t-stat 2.08 and 2.01). In the case of TD3 and VFS-TD3, we can reject the same hypothesis with a confidence level of 90% (t-stat 1.73 and 1.66). The latter results are interesting as they suggest that TD3's implementation tricks to improve performance benefit policy learning as they lead to gradient estimates that are more similar to the VFS ones. Moreover, the right column shows the average MSE of our models computed every $10^5$ step. Similarly to the PPO experiments, we use the actual return for $10^3$ trajectories in the environment to approximate the true value function.

These results follow the trend of Section 4.3, where VFS critics have lower MSE. Interestingly, we also note that TD3 typically achieves a higher gradient estimate correlation with lower variance over

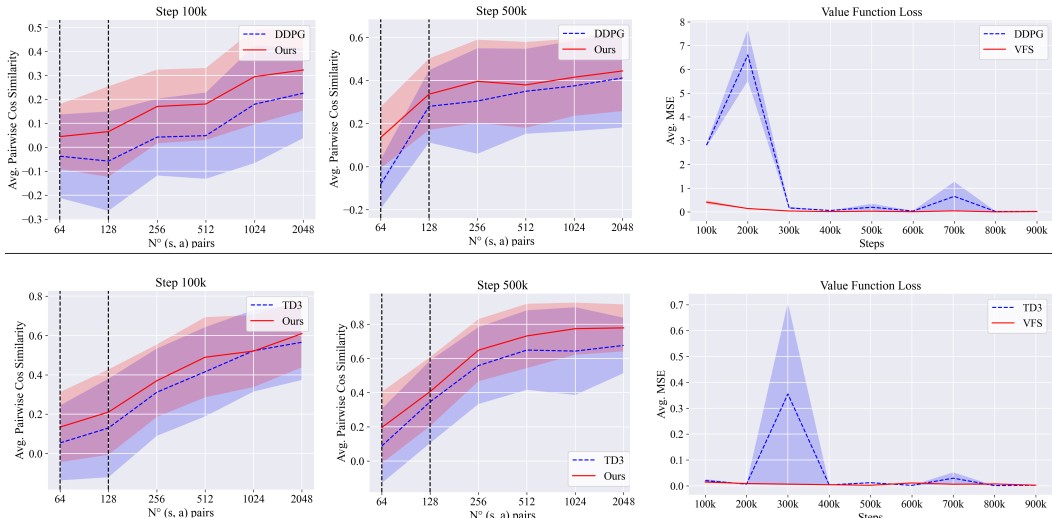

Figure 9: Analysis of gradient estimate and variance (left, middle) in the Hopper task for DDPG and VFS-DDPG in the first row, TD3 and VFS-TD3 in the second row. We measure the average pairwise cosine similarity (y-axis) between 30 gradient estimates which use a growing number of state-action pairs (x-axis). For each algorithm, we use the average over the 25 models collected during the training runs at $10^5$ and $5 \times 10^5$ steps. The higher correlation with a lower variance of VFS-based implementations translates into better performance of the Deep PG algorithm. Average MSE of such models over the training phase (right).

DDPG, which motivates using two value networks to achieve better performance. Our analysis of key Deep PG primitives for DDPG and TD3 confirms the main paper's results.

In addition, we performed the same gradient estimate analysis on two VFS implementations that only employ small or big perturbation operators. Figure 10 shows the results of such an investigation. In particular, VFS with only small perturbations slightly improves over the baseline PPO in the different stages of the training. However, it can not achieve the higher similarity of the VFS-PPO that considers both perturbations. In contrast, VFS with only big perturbations is characterized by a higher correlation in the initial stages of the training. At the same time, its efficacy significantly decreases later in the training phase. Moreover, considering only such big-scale variations typically leads to higher gradient estimate variance. Crucially, this additional gradient estimate analysis confirms the results of Figure 3 and further motivates using our two-scale perturbations to tackle both local and global variations at the same time.

## E  HYPERPARAMETERS

We performed an initial grid search among common hyper-parameters from prior work (Schulman et al., 2017; Lillicrap et al., 2016; Fujimoto et al., 2018; Marchesini et al., 2021a). This evaluation led us to use similar DNN architectures for all the algorithms in the environments, i.e., 2-ReLU layer MLP with 64 hidden units each to model actors and critics for PPO and the same structure but with 256 hidden units for DDPG, TD3. Table 2 reports the other hyperparameters shared across all the experiments. Note that missing parameters are the same as the baseline algorithm (e.g., VFS-PPO values are the same as PPO, and TD3 shares the missing parameters with DDPG). Moreover, all the VFS-based implementations consider the same values for the critic's search.

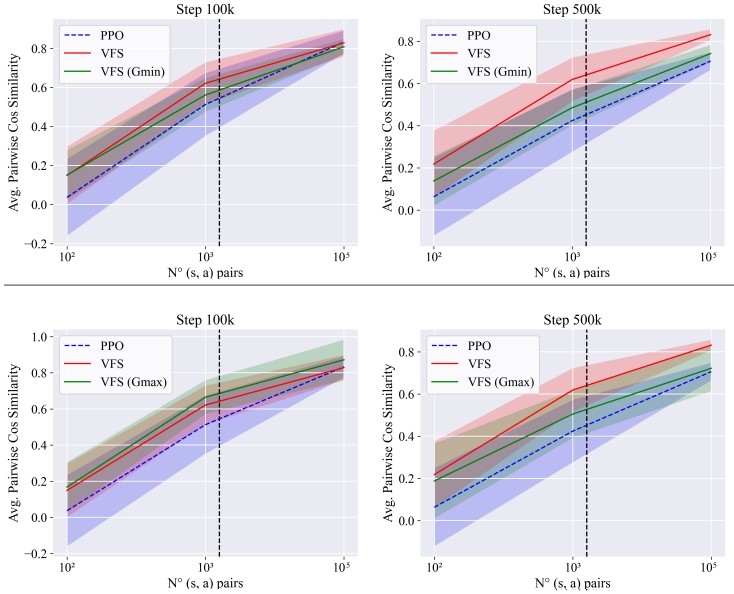

Figure 10: Analysis of gradient estimate and variance in the Hopper task for VFS-PPO that only considers small perturbations (top row) or big ones (bottom row).

Table 2: Hyperparameters for our experiments

| Algorithm | Parameter | Value |
|---|---|---|
| **PPO** | $\epsilon$ clip | 0.2 |
| | $\gamma$ | 0.99 |
| | update epochs | 10 |
| | samples per update | 2048 |
| | mini-batch size | 64 |
| | entropy coefficient | 0.001 |
| | lr | 0.0003/0.0001 |
| **DDPG** | $\gamma$ | 0.99 |
| | buffer size | 1000000 |
| | mini-batch size | 128 |
| | actor lr | 0.0001 |
| | critic lr | 0.001/0.0003 |
| | $\tau$ | 0.005 |
| **TD3** | actor noise clip | 0.5 |
| | delayed actor update | 2 |
| **VFS** | population size | 10 |
| | $\sigma_{\min}$ | 0.000005 |
| | $\sigma_{\max}$ | 0.0005 |
| | $b$ | 2048 |
| | $e_s$ | 2048 steps |

