# OpenReview forum: "Improving Deep Policy Gradients with Value Function Search"
_ICLR.cc/2023/Conference — ICLR 2023 poster_

### Official Review · Reviewer_7FyJ · 2022-10-24

**Confidence:** 4
**Correctness:** 2
**Technical Novelty And Significance:** 3
**Empirical Novelty And Significance:** 3
**Recommendation:** 5

**Clarity, Quality, Novelty And Reproducibility:**

The paper is easy to follow and details on how to add the additional optimization step and hyperparameters are provided. The code is not available.



The idea behind the method is very similar to existing genetic RL algorithms, with the twist of applying the optimization step only to critics and two-scale perturbation of the weights.

**Strength And Weaknesses:**

Overall, the question of how to better approximate values is interesting to the community. The paper aims to tackle this in a minimally invasive way by adapting only the critic's optimization step, which means it is easy to implement and does not require additional data from the environment. The proposal to both locally and globally adapt the weights as part of the evolutionary adaptation of the weights is interesting, and in particular the global step could be related to capacity loss in critics [1] (this review has no affiliation to the authors). The paper is also clearly written and easy to follow.



The papers' main weakness is the experimental evaluation. In particular, no evidence is given that the combination of local and global updates actually improves performance or whether one of them would be sufficient on their own. Since this is framed as one of the main contributions, this makes the paper difficult to evaluate. In addition, some benchmarks seem to be wrongly implemented (especially PPO), since they do not achieve the performance available in open-source RL frameworks (see, for example, [2]). It is also not clear how the methods would compare on an equal computational budget, i.e. if the model-free budget was used to perform more gradient steps of the critic.



[1] "Understanding and Preventing Capacity Loss in Reinforcement Learning", ICLR 2022

[2] https://tianshou.readthedocs.io/en/master/tutorials/benchmark.html

**Summary Of The Paper:**

The paper studies value function approximation in reinforcement learning and proposes to include gradient-free evolutionary training steps when fitting the value function to data from the replay buffer. In particular, after a certain number of training steps the critics weights are perturbed with zero-mean Gaussian noise with two different magnitudes to explore different critic approximations both locally and globally. It then experimentally evaluates several on- and off-policy actor-critic algorithms on the mujoco benchmarks both with and without the additional gradient-free training step for the critic. The experiments show that this can help improve performance.

**Summary Of The Review:**

Overall I think the paper is interesting but, given the similarity to existing genetic RL algorithms, in my opinion the paper does provide sufficient experimental evidence to clear the bar for acceptance at ICLR. In particular, the paper should consider adding an ablation study to confirm how local v.s. global perturbations affect performance, use state-of-the-art implementations for the comparison, and evaluate how more critic gradient steps would perform relative to the model-free training step.

---

### Official Review · Reviewer_qNbH · 2022-10-25

**Confidence:** 4
**Correctness:** 2
**Technical Novelty And Significance:** 4
**Empirical Novelty And Significance:** 3
**Recommendation:** 5

**Clarity, Quality, Novelty And Reproducibility:**

The paper is very clear and easy to follow. The authors present their approach in a straightforward way.
There have been many papers combining evolutionary approaches with RL Usually, the policy is evolved, while the value function (or a population of value functions) is trained by Monte Carlo or Temporal Difference methods. Here instead it is the value function to be evolved in order to better fit the data, so to my knowledge, the method presented is novel. The quality of the paper is good, although some claims should be better supported by experiments (see weaknesses).
The authors report implementation details, making the algorithm easy to reproduce.

**Strength And Weaknesses:**

Strengths:
- The method can be easily implemented and produces little overhead

- The method seems to improve upon the baselines when it is applied

- The paper is clear and easy to follow. The authors provide some ablations indicating important differences when using high or low noise for perturbation

- I appreciate that the authors included Figure 3 in the paper, which confirms the intuition that higher noise might be more helpful at the beginning of training on PPO. Does the same hold for off-policy methods?

Weaknesses:

- The authors show a comparison between the proposed VFS-TD3 and Supe-RL and SUNRISE. This comparison seems not fair unless these two approaches are also built on top of TD3. Is this the case? I believe TD3 itself would outperform the results reported by Supe-RL and SUNRISE, so the comparison does not seem fair to me.
- Some of the reported results for the baselines (PPO, DDPG, TD3) seem lower with respect to the best results in the literature at 2M time steps (see e.g. https://spinningup.openai.com/en/latest/spinningup/bench.html for DDPG and TD3). Why is that the case? What implementation was used for the baselines and how were the hyperparameters chosen? Did the authors tune hyperparameters both for the baseline and their method?
- The authors claim that their method improves value prediction. However, there is no analysis showing value predictions. The value function loss is reported, but a low value function loss does not necessarily imply that the prediction of the value function will be accurate, especially when using off-policy methods with Temporal Difference Learning. Can the authors clarify this or include an experiment that analyzes value prediction?
- The authors claim that their method has reduced variance over the baseline. However, the learning curves in Figure 6 show sometimes that the proposed method has a larger variance. Moreover, while in Ant and HalfCheetah environments VFS-PPO has a larger variance than PPO, Table 1 reports a lower variance for VFS-PPO. Can the authors clarify this?

- Figure 8 does not show significant benefits in cosine similarity between the gradient of the proposed method and the true gradient, as the curves for the proposed methods and the baselines almost completely overlap.

Other questions:

Can the authors give some intuition on what would be the issues in scaling up this method for more complex domains? Could the method be easily applied for e.g. A2C and applied to Atari games?

Are the values in Figure 2 the results of 100 perturbations and then predicting one episode for each, or are they the results of 2 perturbations and then 100 predictions for each perturbation? Since these 100 values have no clear structure, instead of plotting them in a table, the authors might want to consider plotting the distributions of predictions induced by different values of sigma

What is the value of $e_s$, the periodicity hyperparameter?


**Summary Of The Paper:**

The paper introduces an evolutionary approach to train value functions in actor-critic methods. The main motivation for this approach is to provide a better value approximation, a higher correlation between the estimated gradient and the true gradient, and a lower variance. The authors propose to periodically perturb half the population of value functions using low noise and half using higher noise. The intuition for this is that while the small noise can perform a local search around the current value function, using a high noise for some value functions can escape local minima. After perturbation, the value function with the lowest error on a batch of data is selected. The authors apply their method to on-policy (PPO) and off-policy (DDPG, TD3) algorithms. Experiments show that this addition increases sample efficiency over the non-evolutionary baselines, without introducing significant overhead. Further ablations show that at the beginning of training, the high noise population is mostly responsible for value function improvement, while towards the end of training most improvement is due to the small noise perturbation. Moreover, the gradients produced by the presented approach appear to have a higher correlation to the true gradient and the value function loss is lower.

**Summary Of The Review:**

The paper introduces a novel evolutionary approach for evolving value function in actor-critic methods. The method is easy to implement and shows improvement upon on-policy and off-policy baselines. Some of the claims made in the paper should be properly justified and some experimental concerns should be resolved before acceptance.

---

### Official Review · Reviewer_ur4C · 2022-10-25

**Confidence:** 4
**Correctness:** 3
**Technical Novelty And Significance:** 3
**Empirical Novelty And Significance:** 3
**Recommendation:** 6

**Clarity, Quality, Novelty And Reproducibility:**

The paper has great clarity, good quality, and decent novelty.; The reproducibility should be good.

**Strength And Weaknesses:**

Strength
+ Very clear description of the proposed idea, including an illustrative overview, algorithm description, etc;
+ Strong performance when combined with three different base RL algorithms, including PPO, DDPG, and TD3. In all tested domains, the proposed method VFS outperforms the base algorithms w/o VFS significantly;
+ The VFS also achieves a boost in sample efficiency when applied to the base RL algorithms;
+ The performance comparison remains strong when compared against the policy search method Supe-RL and ensemble method SUNRISE

Weaknesses
- In Algorithm 1, splitting the value functions pool into **half** and add $\mathcal{G}_\text{min}$ and  $\mathcal{G}_\text{max}$ seem a bit arbitrary, could you please provide more explanation on this?
- From Figure 7, the population size of value functions seems to have a diminishing return effect. I would like to understand whether the best population size is consistent across tasks. I would suspect it might need more value functions in more complex tasks.

**Summary Of The Paper:**

This paper takes a careful look into how to make value function predictions more accurate, which is often neglected in modern policy gradient methods. The idea is simple, to maintain a population of value functions, and with perturbed value networks to minimize the regression loss. There are a bunch of nice properties of the proposed method including good empirical performance, low additional computation cost.

**Summary Of The Review:**

I'd vote for acceptance of the paper since its simplicity and nice empirical performance.

---

### Official Review · Reviewer_KmE6 · 2022-11-01

**Confidence:** 4
**Correctness:** 3
**Technical Novelty And Significance:** 2
**Empirical Novelty And Significance:** 3
**Recommendation:** 5

**Clarity, Quality, Novelty And Reproducibility:**

**Clarity**

The overall writing is very clear. One can clearly understand the main arguments of the paper, from both the experimental results and the description of the method.

**Quality and Reproducibility**

This paper proposes VFS and explains the high-level idea of the VFS training process. However, as mentioned above, there is a lack of explanation on how VFS obtain the V^{\pi}(s) in the MSE loss and how VFS ensures that it can escape from local optima (via the use of noise with larger variance)

**Novelty**

This paper provides the first population-based method specifically for learning critics for deep PG. That said, VFS still falls within the concept of population-based search for PG, whose benefits have been quite extensively studied in the literature. The idea of VFS does not appear very novel.


**Strength And Weaknesses:**

**Strengths**

- The motivation for enhancing the sample efficiency by improving the estimation of value function is convincing: (i) ensemble methods require more additional inference time and (ii) prior works require multiple training models or additional environment interactions.
- This paper proposes a gradient-free population-based critic learning approach which does not require complex gradient computation and serves as a computationally inexpensive approach to enhance the accuracy of value estimates.
- An empirical evaluation and an ablation study that indicates the effectiveness of the proposed VFS in critic training: (i) The value function loss and similarity of gradient estimates in Figure 4 shows the influence of VFS. (ii) The experimental results in Table 1 show that the VFS-based Deep PG algorithms outperforms the primitive algorithm in terms of average return.

**Weaknesses**

My concerns are mainly two-fold:
- A (nearly) true V^{\pi} needed in the loss function: To enable gradient-free critic updates, VFS uses the simple MSE as the loss function (cf. Eq (7)), which measures the difference between the true V^{\pi} and the perturbed critics. This is reasonable if a nearly true V^{\pi} is available. While such V^{\pi} can be obtained by Monte Carlo methods with a large number of trajectories, this could lead to sample inefficiency, which contradicts one of the motivations of this paper. On the other hand, if such V^{\pi} is inaccurate, the MSE loss could lead to significant bias in the learned perturbed critics. As the paper does not clearly specify how VFS obtains V^{\pi} in its implementation, it is not immediately clear how to interpret the empirical performance improvement provided in Table 1 and the comparison of MSE in Figure 4 (since a lower MSE does not necessarily indicate a more accurate critic if V^{\pi} is inaccurate).  However, this is typically not the case as one would not have a sufficiently accurate estimate of V^{\pi} to begin with (otherwise critic learning would not be needed).
- Given the several existing prior works on population-based methods for deep PG (e.g., [Khadka & Tumer, 2018; Marchesini & Farinelli, 2020; Marchesini et al., 2021; Sigaud, 2022] and references therein), the idea of using a population-based approach in VFS is not very novel.

**Additional questions**
- The result in Figure 2 appears quite straightforward and does not seem to provide much insight into the performance of VFS. By contrast, experiments like Figure 7 and Figure 6 are actually more informative and would be helpful in the main text. Is there any specific argument that the author(s) would like to make through Figure 2?
- How to calculate $V_\pi(s)$ in Eq (7)? How to strike a balance between getting an accurate $V_\pi(s)$ and sampling efficiency? Do other baselines also use true value functions to assist the update?
- In Algorithm 1, VFS utilizes a two-scale approach to perturb the parameter $\phi$ which is the best fit of the true value function. What would happen if one either uses only noise with small variance or only noise with high variance? Moreover, why is it needed to combine both types of noise? More ablation studies are surely needed.



**Summary Of The Paper:**

This paper aims to enhance the sample efficiency of Deep PG algorithms by improving the learning of value functions. To this end, this paper proposes Value Function Search (VFS), which instantiates a population of perturbed critics and uses two-scale perturbation noise to make good use of two advantages: (1) noise with small variance is to find a better set of parameters within a local search. (2) noise with big variance is to escape from local optima by exploring. The experiments show that the proposed framework outperforms the baselines of VFS-based algorithms. Ablation studies show how VFS affects the training phase and how VFS can enhance the Deep PG primitives.

**Summary Of The Review:**

This paper proposes a gradient-free population-based method to update the value network. The experiments show that the proposed framework outperforms the baseline methods.
That said, here are the two major concerns: (1) If a nearly true value function (for calculating MSE loss and finding the best fit within the perturbed critics) can be obtained at a low cost during the training process, then it appears sufficient to directly use the true value function to assist Deep PG in the policy updates. Additional ablation studies are needed to confirm the benefits of value function search. (2) While this paper's method of perturbing the value network using two-scale noise terms is interesting, the overall design is quite similar to other population-based methods. The idea of VFS does not appear very novel.

---

### Decision · Program_Chairs · 2023-01-20

**Decision:**

Accept: poster

**Justification For Why Not Higher Score:**

Remaining concerns regarding novelty.

**Justification For Why Not Lower Score:**

Solid contribution that demonstrates consistent gains over common baselines. Rebuttal that adequately addressed reviewer's concerns.

**Metareview: Summary, Strengths And Weaknesses:**

I thank the authors for their submission and active engagement during the discussion period.  This is a borderline paper. On the positive side, reviewers remarked the convincing motivation [KmE6], a thorough empirical evaluation with ablations [KmE6], the clarity of the paper [KmE6,qNbH], convincing improvements over baselines [ur4C,qNbH]. On the negative side, reviewers were concerned about limited novelty [KmE6], and the results of the baselines compared to results reported elsewhere [qNbH,7FyJ]. In my view the authors have adequately addressed major concerns voiced by the reviewers, while the reviewers don't seem to have updated their scores based on the rebuttal. I believe the strengths outweigh the weaknesses of this paper and therefore recommend acceptance. I strongly encourage the authors to use the feedback by the reviewers to improve their paper.

**Note From Pc:**

if the above contains the word "oral" or "spotlight" please see: "oral" presentation means -> notable-top-5% and "spotlight" means -> notable-top-25%. As stated in our emails, we are disassociating presentation type from AC recommendations